# Robust Filtering Techniques for RTK Positioning in Harsh Propagation Environments

**DOI:** 10.3390/s21041250

**Published:** 2021-02-10

**Authors:** Daniel Medina, Haoqing Li, Jordi Vilà-Valls, Pau Closas

**Affiliations:** 1Institute of Communications and Navigation, German Aerospace Center (DLR), 17235 Neustrelitz, Germany; 2Electrical and Computer Engineering Department, Northeastern University, Boston, MA 02115, USA; li.haoq@northeastern.edu (H.L.); closas@ece.neu.edu (P.C.); 3Institut Supérieur de l’Aéronautique et de l’Espace (ISAE-SUPAERO), University of Toulouse, 31055 Toulouse, France; jordi.vila-valls@isae-supaero.fr

**Keywords:** GNSS, RTK, precise positioning, multipath, Kalman filtering, robust filtering

## Abstract

Global navigation satellite systems (GNSSs) play a key role in intelligent transportation systems such as autonomous driving or unmanned systems navigation. In such applications, it is fundamental to ensure a reliable precise positioning solution able to operate in harsh propagation conditions such as urban environments and under multipath and other disturbances. Exploiting carrier phase observations allows for precise positioning solutions at the complexity cost of resolving integer phase ambiguities, a procedure that is particularly affected by non-nominal conditions. This limits the applicability of conventional filtering techniques in challenging scenarios, and new robust solutions must be accounted for. This contribution deals with real-time kinematic (RTK) positioning and the design of robust filtering solutions for the associated mixed integer- and real-valued estimation problem. Families of Kalman filter (KF) approaches based on robust statistics and variational inference are explored, such as the generalized M-based KF or the variational-based KF, aiming to mitigate the impact of outliers or non-nominal measurement behaviors. The performance assessment under harsh propagation conditions is realized using a simulated scenario and real data from a measurement campaign. The proposed robust filtering solutions are shown to offer excellent resilience against outlying observations, with the variational-based KF showcasing the overall best performance in terms of Gaussian efficiency and robustness.

## 1. Introduction

Modern intelligent transportation systems and other safety-critical applications require reliable, continuous, and precise positioning, navigation, and timing (PNT) information for their successful operation and implantation in the market. Global navigation satellite systems (GNSSs) constitute the backbone and main information supplier of PNT data [1,2,3], and this dependence can only grow in the future [4]. Despite offering a fairly good open sky positioning performance, standard code-based GNSS techniques can solely achieve metre-level accuracy, which is insufficient for a plethora of applications that require high precision. Instead, the use of carrier phase measurements enables us to reach decimeter to centimeter level accuracies. Two distinct techniques exploit carrier phase observations, namely precise point positioning (PPP) [5] and real-time kinematic (RTK) [6]. In this work, we focus on RTK, a relative positioning procedure for which the position of a moving receiver is determined with respect to (w.r.t.) a nearby base station of known geolocation. Although accuracies of up to the centimeter level can be reached almost immediately, RTK requires a dense network of reference stations and a sufficient communication channel [7]. This work deals with RTK positioning and its challenges in urban scenarios, where multipath effects can heavily deteriorate the navigation solution.

With respect to standard GNSS code solutions, RTK involves a more complex estimation process [8], since a vector of double-difference integer ambiguities is estimated along with the target receiver position. Given the integer nature of such ambiguities, the most common solution consists of a three step approach [9]. These are applied sequentially: (i) float solution determination, (ii) integer ambiguity resolution, and (iii) state reconstruction or fixed solution estimation. In the initial float estimation step, either a least-squares (LS) or KF-type method is considered, where the ambiguities and other unknowns are estimated as real numbers (which is the reason why the results are often referred to as the *float solution*). Then, the estimated float ambiguities and the corresponding covariance matrix are used to obtain the integer ambiguity estimates. Finally, the computed integer ambiguities are used to improve the positioning solution. Such an estimate is typically implemented, again, within the LS framework in order to obtain the so-called *fixed solution*.

The standard RTK three-step approach described above is known to provide a good estimation performance under nominal line-of-sight (LOS) conditions. Unfortunately, this is not the case for urban environments, typically affected by multipath and non-line-of-sight (NLOS) propagation conditions. Indeed, the phase observables’ quality is directly linked to the corresponding code-based delay observables. In the maximum likelihood sense, the phase estimate is given by the argument of the cross-ambiguity function evaluated at the corresponding delay/Doppler estimates. Therefore, a worse delay estimate (i.e., induced by multipath conditions) has a direct impact on the phase estimate, and this drives the phase observable achievable performance/quality [10]. Because of the very low ultimate positioning precision exploiting phase measurements, the impact of worse phase observables is much more critical than that for a code-based solution. Therefore, these harsh propagation conditions are the main source of errors for precise navigation, since the locality of the effect prevents augmentation systems from assisting meaningfully against such channel effects. As a consequence, counter measuring signal reflections can be performed locally at the receiver in either the baseband processing (i.e., in the calculation of observables) or the navigation side (i.e., when producing the PVT solution).

The use of robust approaches based on either robust statistics or on variational inference has been extensively applied for different purposes within the context of GNSS-based positioning. Table 1 summarizes the state-of-art on robust estimation for GNSS applications. In this work, we explore the use of robust solutions in order to improve RTK-based precise positioning at the last receiver stage (i.e., navigation solution), enhancing the usual RTK pipeline to provide robustness to local effects. Several robust Kalman filters (KF) are considered, providing a meaningful comparison of existing techniques and a new robust RTK approach that is resilient to, for instance, multipath conditions. The key point is to improve the performance of float solution estimation, where multipath-contaminated satellite links are detected, enabling the guidance of an integer ambiguity search and enhancing its success ratio, being the most critical point in harsh conditions. To achieve such a goal, we propose to use and analyse a set of different robust KFs with outlier detection and rejection capabilities: (i) four robust information filters (RIFs) using different robust cost functions or a 3σ rejection rule, (ii) a generalized M-estimate robust regression KF, and (iii) two variational Bayesian inference-based robust KFs.

The different robust filter performances are assessed using both synthetic and real data on challenging propagation environments. The simulated scenario allows us to characterize the estimators in terms of Gaussian efficiency, i.e., how good their performance is under nominal noise conditions, and robustness based on Monte Carlo experimentation. The real data was collected on board of a vessel that travelled through a river with several bridges and a waterway lock and then was affected by severe multipath and NLOS conditions. The evaluation is comprises of the positioning accuracy of the robust RTK solution for the different methods, alongside the corresponding fixing rates. The fixing rate defines the probability of finding the correct set of integer ambiguities based on validation tests from the Integer Aperture (IA) framework to decide whether the estimated set of integer ambiguities can be considered sufficiently more likely than the best counter-hypothesis [31]. The reported results show the improved robustness, both in terms of positioning precision and fixing rate, when adding a robust KF in the RTK pipeline, providing a promising solution for real-time robust RTK in harsh propagation conditions.

## 2. Background on GNSS RTK Precise Positioning

RTK is a relative positioning procedure, where the unknown position of a moving rover station is determined with respect to a stationary base station of known coordinates. Figure 1 (left) illustrates the working principle. Due to observation single-differencing, the satellites’ position and clock errors are completely eliminated while the satellite orbit error and the atmosphere-related delays can be reduced significantly depending on the distance between the base and rover positions ($p_{B}$ and $p_{R}$ respectively). Then, a pivot satellite (hereinafter referred to using the superscript *r*) is chosen for double-differencing, cancelling the delays from the receivers’ clock offsets. Let us consider n+1 GNSS satellites (i.e., *n* satellites plus the pivot) tracked simultaneously by the rover and base receivers. The observations for code and phase pseudoranges from the *i*th satellite are given by the following:(1)$$\rho^{i}=∥p^{i}-p∥+I^{i}+T^{i}+c\left(dt-dt^{i}\right)+\varepsilon^{i}$$
(2)$$\Phi^{i}=∥p^{i}-p∥-I^{i}+T^{i}+c\left(dt-dt^{i}\right)+\lambda N^{i}+\epsilon^{i}$$
where

$\rho^{i}$ and $\Phi^{i}$ are the code and phase observations (m), respectively;$p^{i}$ and p are the positions of the *i*th satellite and the GNSS receiver, respectively;$I^{i}$ is the ionospheric error (m);$T^{i}$ is the tropospheric error (m);*c* is the speed of light (299,792,458m/s);$dt^{i}$ and dt are the satellite and receiver clock offsets (s), respectively;λ is the carrier phase wavelength (m);$N^{i}$ is an unknown number of cycles between the receiver and the satellite; and$\varepsilon^{i}$ and $\epsilon^{i}$ are the remaining unmodelled errors for the code and phase observations, respectively.

Then, the double-difference (DD) code and phase observations results are
(3)$$\begin{array}{cc} DD\rho^{i} & ≜\;\rho_{R}^{i}-\rho_{B}^{i}-\left(\rho_{R}^{r}-\rho_{B}^{r}\right)\\ \; & =∥p^{i}-p_{R}{∥}^{2}-∥p^{i}-p_{B}{∥}^{2}-∥p^{r}-p_{R}{∥}^{2}+{∥p^{r}-p_{B}∥}^{2}+\varepsilon_{R,B}^{i,r}, \end{array}$$
(4)$$\begin{array}{cc} DD\Phi^{i} & ≜\Phi_{R}^{i}-\Phi_{B}^{i}-\left(\Phi_{R}^{r}-\Phi_{B}^{r}\right)\\ \; & =∥p^{i}-p_{R}{∥}^{2}-∥p^{i}-p_{B}{∥}^{2}-∥p^{r}-p_{R}{∥}^{2}+{∥p^{r}-p_{B}∥}^{2}+\lambda a^{i}+\epsilon_{j,b}^{i,r}, \end{array}$$
where the first lines in Equations (Equation 3) and (Equation 4) correspond to the actual observation combination and the second lines relate to the observation model (the relationship between the observations and the unknown parameters). The vector y gathers the DD observations as
(5)$$y={\left[\begin{array}{c} DD\Phi^{1},\cdots ,DD\Phi^{n},DD\rho^{1},\cdots ,DD\rho^{n} \end{array}\right]}^{⊤},\;y\in R^{2n,1}.$$

The resulting positioning observation model is typically linearized and expressed as
(6)y=Aa+Bb+η,
with $a\in Z^{n}$ is the vector of DD ambiguities, b is the baseline vector between rover and base positions $b=p_{R}-p_{B}$, and η is the zero-mean noise term with covariance R such that $\eta ∼N(0_{2n,1},R)$. The design matrices are described by
(7)$$\begin{array}{c} A=\left[\begin{array}{c} \lambda I_{n}\\ 0_{n} \end{array}\right],\;B=\left[\begin{array}{c} G\\ G \end{array}\right],\;G=\left[\begin{array}{c} -{\left(u^{1}-u^{r}\right)}^{⊤}\\ \vdots \\ -{\left(u^{n}-u^{r}\right)}^{⊤} \end{array}\right], \end{array}$$
where $I_{n}$ and $0_{n}$ denote, respectively, the *n*-dimensional unit and null matrices. The geometry matrix G is composed of the satellite steering line-of-sight vectors w.r.t. the base position $u^{i}$. The observation covariance matrix R is generally defined based on stochastic models dependent on satellite elevation and/or carrier-to-noise density ratio [32,33]. Next, solving the RTK positioning problem in (Equation 6) is discussed for snapshot (memoryless) and recursive estimation.

### 2.1. A Least-Squares Approach

The RTK positioning problem is also known under the mixed model estimation problem, given that both real- and integer-valued parameters are to be estimated. The related optimization problem is formulated as
(8)$$\left\{a,b\right\}=\arg\min_{\begin{array}{c} a\in Z^{n}\\ b\in R^{3} \end{array}}{∥y-Aa-Bb∥}_{R}^{2}.$$

Provided that no analytical solution exists for the minimization (Equation 8), its decomposition into a series of three consecutive LS adjustments was proposed [9], written as
(9)$$\min_{\begin{array}{c} a\in Z^{n}\\ b\in R^{3} \end{array}}{∥y-Aa-Bb∥}_{R}^{2}=\underset{\mathrm{float}\;\mathrm{solution}}{\underset{︸}{\min_{\begin{array}{c} \hat{a}\in R^{n}\\ \hat{b}\in R^{3} \end{array}}{∥y-A\hat{a}-B\hat{b}∥}_{R}^{2}}}+\underset{\mathrm{ILS}}{\underset{︸}{\min_{a\in Z^{n}}{∥\hat{a}-a∥}_{P_{\hat{a}\hat{a}}}^{2}}}+\underset{\mathrm{fix}\;\mathrm{solution}}{\underset{︸}{\min_{b\in R^{3}}{∥\hat{b}|a-b∥}_{P_{\hat{b}|a}}^{2}}}.$$

The overall three-step procedure is illustrated in Figure 1 (right). The first term corresponds to the so-called *float estimation*, an LS adjustment where the integer constraints on a are disregarded. The distribution of the float solution is
(10)$$\left[\begin{array}{c} \hat{a}\\ \hat{b} \end{array}\right]∼N\left(\left[\begin{array}{c} \hat{a}\\ \hat{b} \end{array}\right],\left[\begin{array}{cc} P_{\hat{a}\hat{a}} & P_{\hat{a}\hat{b}}\\ P_{\hat{b}\hat{a}} & P_{\hat{b}\hat{b}} \end{array}\right]\right),$$
where P is the associated covariance matrix that gathers the uncertainty of the float estimates $\hat{a}$ and $\hat{b}$ and their cross-correlation.

The second step, known as integer ambiguity resolution (IAR), incorporates the integer constraints into the previously estimated float ambiguities. Thus, real-to-integer mapping $I:R^{n}\to Z^{n}$ is the process that assigns the float ambiguity vector to an integer one $a\in Z^{n}$, such that $a=I\left(\hat{a}\right)$. While different estimators have been proposed, the integer least squares (ILS) provides the optimal performance [34,35,36]. Integer estimators that include a validation step belong to the Integer Aperture (IA) framework [37]. The latter allows us to discern whether an integer solution can be accepted based on a predefined failure rate [38].

Finally, the third minimization problem improves the positioning estimate with knowledge of the integer ambiguities a, driving a highly accurate position solution, denoted as the *fixed solution*. The mean b and covariance $P_{bb}$ of the fixed estimate are based on the projection of the estimated integer ambiguities on the position domain as
(11)$$\begin{array}{cc} b & =\hat{b}-P_{\hat{b}\hat{a}}P_{\hat{a}\hat{a}}^{-1}\left(\hat{a}-a\right), \end{array}$$
(12)$$\begin{array}{cc} P_{bb} & =P_{\hat{b}\hat{b}}-P_{\hat{b}\hat{a}}P_{\hat{a}\hat{a}}^{-1}P_{\hat{a}\hat{b}}. \end{array}$$

A relevant remark is that, whenever the estimated integer ambiguities do not match the true ones, the fixed solution will be biased. The precision of the solution improves only when the correct ambiguities are estimated.

### 2.2. A Kalman Filter Approach

To exploit recursive estimation in navigation problems, the KF and its nonlinear extensions, i.e., the Extended and Sigma-Point Gaussian Filters (EKF and SPGF, respectively), are typically applied. This work focuses on the EKF (notice that a typical RTK observation model in Equation (Equation 6) is implicitly a linearized version) and its application to RTK, leading to the approximation of the following marginal posterior distribution:(13)$$\begin{array}{c} p\left(x_{t}|y_{1:t}\right)=N\left(x_{t};x_{t|t},P_{t|t}\right),\;\mathrm{with}\;x_{t}={\left[\begin{array}{c} b_{t}^{⊤},v_{t}^{⊤},a_{t}^{⊤} \end{array}\right]}^{⊤}, \end{array}$$
where $x_{t}$ is the random variable that we aim to estimate, and $x_{t|t}$ and $P_{t|t}$ are the estimated posterior mean and covariance, respectively. The time evolution of the state estimate is dictated by the process and measurement functions, f(·) and h(·), respectively, also known as the prediction and correction models:(14)$$\begin{array}{cc} \; & x_{t}=f\left(x_{t-1}\right)+w_{t}, \end{array}$$(15)$$\begin{array}{cc} \; & y_{t}=h\left(x_{t}\right)+\eta_{t}, \end{array}$$
where the process and observation noises are assumed to follow normal distributions $w_{t}∼N\left(0,Q_{t}\right)$, $\eta ∼N(0,R_{t})$. With regards to the dynamic process, the rover is generally assumed to follow a constant velocity model [39] (unless the vehicle is equipped with inertial sensing capabilities) while the ambiguities are assumed to remain constant over time as long as no cycle slip or signal reacquisition occurs.

The observation model integrates the DD observations in (Equation 5). As for the afore-described LS approach, KF-based RTK resorts to the mixed model parameter estimation. Adding the discrete time dependency and the velocity estimation, the observation model (Equation 6) can be expressed as
(16)$$y_{t}=\underset{H_{t}}{\underset{︸}{\left[\begin{array}{ccc} G_{t} & 0_{n,3} & \lambda I_{n}\\ G_{t} & 0_{n,3} & 0_{n} \end{array}\right]}}x_{t}+\eta_{t}=A_{t}a_{t}+\underset{{\tilde{B}}_{t}}{\underset{︸}{\left[\begin{array}{cc} G_{t} & 0_{n,3}\\ G_{t} & 0_{n,3} \end{array}\right]}}\underset{{\tilde{b}}_{t}}{\underset{︸}{\left[\begin{array}{c} b_{t}\\ v_{t} \end{array}\right]}}+\eta_{t}.$$

The classical KF correction procedure follows the well-known expression
(17)$$\begin{array}{cc} K_{t} & =P_{t|t-1}H_{t}^{⊤}{\left(H_{t}P_{t|t-1}H_{t}^{⊤}+R_{t}\right)}^{-1}, \end{array}$$
(18)$$\begin{array}{cc} x_{t|t} & =x_{t|t-1}+K_{t}\left(y_{t}-h\left(x_{t|t-1}\right)\right), \end{array}$$
(19)$$\begin{array}{cc} P_{t|t} & =P_{t|t-1}-K_{t}H_{t}P_{t|t-1}, \end{array}$$
or equivalent on its LS adjustment form [21]: (20)$$\begin{array}{cc} x_{t|t} & ={\left({\tilde{H}}_{t}^{⊤}{\tilde{R}}_{t}^{-1}{\tilde{H}}_{t}\right)}^{†}{\tilde{H}}_{t}^{⊤}{\tilde{R}}_{t}^{-1}{\tilde{y}}_{t}, \end{array}$$(21)$$\begin{array}{cc} P_{t|t} & ={\left({\tilde{H}}_{t}^{⊤}{\tilde{R}}_{t}^{-1}{\tilde{H}}_{t}\right)}^{†}, \end{array}$$
where the $\;\tilde{}\;$ is used to refer to augmented vectors and matrices and ${\left(\;\cdot \;\right)}^{†}$ is the Moore–Penrose inverse. Hence, the augmented observation vector ${\tilde{y}}_{t}$ includes the predicted state $x_{t|t-1}$, and the measurements’ covariance matrix and the observation Jacobian matrix $H_{t}$ change consequently as follows:(22)$${\tilde{y}}_{t}=\left[\begin{array}{c} y_{t}\\ x_{t|t-1} \end{array}\right],\;{\tilde{R}}_{t}=\left[\begin{array}{cc} R_{t} & \\ & P_{t|t-1} \end{array}\right],\;{\tilde{H}}_{t}=\left[\begin{array}{c} H_{t}\\ I \end{array}\right].$$

When dealing with recursive RTK, using the augmented model in (Equation 20) and (21) is more convenient than the classical formulation (Equation 17)–(19), since we can resort to a minimization problem which resembles (Equation 9):(23)$$\left\{a_{t},{\tilde{b}}_{t}\right\}=\arg\min_{a_{t}\in Z^{n},{\tilde{b}}_{t}\in R^{6}}{∥{\tilde{y}}_{t}-A_{t}a_{t}-{\tilde{B}}_{t}{\tilde{b}}_{t}∥}_{{\tilde{R}}_{t}}^{2}.$$

The optimization problem (Equation 23), which includes prior knowledge on the state estimate, is solved following exactly the same three-step decomposition in (Equation 9), as described in Section 2.1. While still sensitive to wrong integer ambiguity estimations, recursive RTK exploits knowledge from previously estimated ambiguities and provides an overall better navigation solution.

## 3. Robust Kalman Filtering Approaches

The standard RTK EKF-ILS solution is close to optimal (i.e., the EKF is an approximation to the nonlinear filtering problem) under nominal LOS propagation conditions. However, its performance is expected to clearly degrade if affected by multipath/NLOS conditions. From the maximum a posteriori estimation point-of-view, the KF update stage relates to the following minimization problem [40]:(24)$$x_{t}=\arg\min_{x_{t}}\left(∥x_{t}-x_{t|t-1}{∥}_{P_{t|t-1}}^{2}+∥y_{t}-h\left(x_{t}\right){∥}_{R_{t}}^{2}\right).$$

Whenever the noise distribution is perfectly known and normally distributed, EKF provides a quasi-optimal solution—later referred to as an ideal EKF in Section 4. In contrast, real-world situations often imply unknown noise statistics and the presence of spurious observations. In such scenarios, EKF operates suboptimally and the influence of the outliers can completely spoil the estimation.

Despite the mixture of integer- and real-valued estimation in RTK, robust integer estimators are not known. Thus, the key to robustifying RTK lies in improving the performance of the float solution estimation. To do so, multipath-contaminated satellite links are detected and their impact on the estimation is mitigated, enabling the guidance of the integer ambiguity search and enhancing its success ratio. Such an improvement is obtained by resorting to different robust KF techniques, which replace the standard first-stage EKF in Figure 1 (right). In the sequel, two categories of filters are discussed: solutions based on the robust statistical framework and approaches based on variational Bayesian inference techniques.

### 3.1. Robust Statistics-Based Filtering

Robust statistics provide alternative loss functions, which appropriately penalize outliers in the measurements in optimization problems [41,42]. For instance, filtering problems such as (Equation 24) are instead expressed as
(25)$$x_{t}=\arg\min_{x_{t}}\left(∥x_{t}-x_{t|t-1}{∥}_{P_{t|t-1}}^{2}+∥y_{t}-h\left(x_{t}\right){∥}_{{\bar{R}}_{t}}^{2}\right),$$
where ${\bar{R}}_{t}$ is the estimated covariance matrix of the observations based on certain weighting functions and formulated as
(26)$${\bar{R}}_{t}=R_{t}^{1/2}W_{y}^{-1}R_{t}^{⊤/2},$$
where $R_{t}^{1/2}$ is obtained from the Cholesky factorization of $R_{t}$ and $W_{y}$ is the weighting matrix given by
(27)$$W_{y}=\mathrm{diag}\left[w\left(R_{t}^{-1/2}\left(y_{t}-h\left(x_{t}\right)\right)\right)\right],$$
where w(·) is a certain weighting function derived from its robust score function ψ(·). A wide family of score functions exists and are classified in *monotone* or *redescending* according to their shape. The idea is to down-weight or nullify (for redescending functions) the effect of observations not fitting the underlying noise model. Figure 2 illustrates some well-known score and weighting functions, such as the $\ell_{2}$ norm (the standard KF score function), the monotone Huber ([43] Equation (1.21)), the bisquare Tukey ([43] Equation (1.23)), and the Institute of Geodesy and Geoinformation (IGG) ([44] Equation (Equation 7)) functions. These functions present a tuning parameter that controls efficiency in the normal case (i.e., when all the observations are normally distributed) or, in other words, their sensitivity in detecting outliers. An interested reader might refer to [19,42] for more details on classical and modern robust functions.

Within the robust statistics-based filtering solutions, one can distinguish between resilience against outliers in the correction step (for robust information filters) or against outliers in both the prediction and correction steps (for the generalized M-estimator KF). Next, two approaches are described:**Robust Information Filters (RIF)**. The information filter (IF) is an algebraically equivalent form of the KF, where instead of the state vector and covariance matrix, the filter propagates the so-called information vector, $z_{t}=P_{t}^{-1}x_{t}$, and information matrix, $Z_{t}=P_{t}^{-1}$. Instead of the standard IF recursion [45,46], the framework of RIF iteratively performs the following procedure until the convergence is reached:
(28)$$\begin{array}{cc} z_{t|t} & =z_{t|t-1}+H_{t}^{⊤}R_{t}^{-⊤/2}\;W_{y}\;R_{t}^{-1/2}H_{t}\left(y_{t}-h\left(x_{t|t-1}\right)+H_{t}x_{t|t-1}\right) \end{array}$$(29)$$\begin{array}{cc} Z_{t|t} & =Z_{t|t-1}+H_{t}^{⊤}R_{t}^{-⊤/2}\;W_{y}\;R_{t}^{-1/2}H_{t}. \end{array}$$This formulation is particularly interesting in the context of robust filtering. Indeed, the use of redescending loss functions (where the weight functions go to zero), may cause numerical issues within the standard robust regression KF (i.e., the weight functions must be inverted). Using the IF formulation, it is possible to avoid these numerical issues and to exploit redescending cost functions. The generic robust information filter (RIF) formulation for nonlinear/Gaussian state-space models was proposed in [44]. During Section 4. Four representative robust functions are used as RIFs: (i) a RIF using a Huber function, (ii) a RIF using a Tukey function, (iii) a RIF using IGG weighting, and *iv)* a RIF using a simple 3σ rejection rule ([43] Chapter 7).**Generalized M-estimator KF (GMKF)**. The LS form of the KF correction in (Equation 20) and (21) is exploited by the GMKF to offer robustness against outlying observations and innovation outliers from the prediction step. For RIF, GMKF consists of an iterative process until convergence of the solution is reached:
(30)$$\begin{array}{cc} x_{t|t} & ={\left({\tilde{H}}_{t}^{⊤}{\tilde{R}}_{t}^{-⊤/2}\;{\tilde{W}}_{\tilde{y}}{\tilde{R}}_{t}^{-1/2}\;{\tilde{H}}_{t}\right)}^{†}{\tilde{H}}_{t}^{⊤}{\tilde{R}}_{t}^{-⊤/2}\;{\tilde{W}}_{\tilde{y}}{\tilde{R}}_{t}^{-1/2}\;{\tilde{y}}_{t}, \end{array}$$
where ${\tilde{W}}_{\tilde{y}}$ is estimated using Equation (Equation 27) over the augmented vector of observations $\tilde{y}$. Once the convergence is reached, the covariance matrix of the associated estimate is
(31)$$\begin{array}{cc} P_{t|t} & ={\left({\tilde{H}}_{t}^{⊤}{\tilde{R}}_{t}^{-⊤/2}\;{\tilde{W}}_{\tilde{y}}{\tilde{R}}_{t}^{-1/2}\;{\tilde{H}}_{t}\right)}^{†}. \end{array}$$Unlike RIF, GMKF involves realizing the inverse operations over the weighting matrix, leading to potential numerical issues upon the use of redescending functions. Moreover, since more “observations” (the actual observations and the predicted state estimate) are weighted, the search space for the robust mechanism grows and leads to a slight poorer performance for the case when outliers are present only during the correction stage. A case disregarded in this work relates to protection against structural errors (i.e., wrong observation and/or prediction models), for which a similar GMKF was introduced in [47].

### 3.2. Variational-Based Filtering

A fundamentally different approach with respect to robust statistics-based filtering is to resort to variational Bayesian (VB) inference techniques [48,49] to detect and mitigate the impact of outliers. This was originally proposed in [50] and further extended and analyzed in [29,30]. The underlying idea is to recursively estimate the probability of having outliers and to then exclude the corrupted observations. The state-space formulation of the problem is as follows: (32)$$\begin{array}{cc} x_{t} & =f\left(x_{t-1}\right)+\epsilon_{t} \end{array}$$(33)$$\begin{array}{cc} y_{t} & =\left\{\begin{array}{cc} h\left(x_{t}\right)+\eta_{t} & \;,\;\mathrm{under}\;M_{0}\\ h\left(x_{t}\right)+\eta_{t}+o_{t} & \;,\;\mathrm{under}\;M \end{array}\right. \end{array}$$
where we have an observation model under nominal conditions ($M_{0}$) and one with additional outliers under non-nominal conditions (M). Again, $x_{t}$ is the state estimate; $y_{t}$ is the vector of observations; $\epsilon_{t}$ is the process noise; $\eta_{t}∼N\left(0,R_{t}\right)$ is the measurement noise; $o_{t}\in R^{n_{y}}$ represents outliers on some or all observations in $y_{t}$ of an unknown distribution; and $n_{y}=2n$ is the total number of double-difference code and carrier-phase measurements. Taking into account this state-space model, the methods in [29,30,50] propose to use an outlier indicator vector $\zeta_{t}\in Z={\left\{0,1\right\}}^{n}$, such that $\zeta_{t}^{\left(i\right)}=0$ if there is an outlier on the *i*th (corrupted) element of $y_{t}$, i.e., $y_{t}^{\left(i\right)}$, and $\zeta_{t}^{\left(i\right)}=1$ if the *i*th element is otherwise clean (not corrupted). In the latter, the nominal Gaussian modeling would be applied, whereas in the former, the wrong information brought by $y_{t}^{\left(i\right)}$ must be down-weighted. Then, instead of approximating the usual posterior distribution $p\left(x_{t}|y_{t}\right)$, these methods provide an approximation of the augmented posterior $p\left(x_{t},\zeta_{t}|y_{t}\right)$. Such an approximation is obtained by resorting to VB inference techniques. We can distinguish two cases of interest:**Scalar VB Kalman filter (S-VKF)**: a scalar outlier indicator $\zeta_{t}$ is used for all observations gathered in $y_{t}$ [50]. In the problem at hand, this may have strong implications because, if a single outlier is detected, then the complete set of observations is disregarded. The following model is considered:
(34)$$p\left(y_{t}|x_{t},\zeta_{t}\right)=N{\left(y_{t};h\left(x_{t}\right),R_{t}\right)}^{\zeta_{t}}$$
where $\zeta_{t}$ is an outlier indicator that assists in detecting and mitigating the impact of outliers. Particularly, when $\zeta_{t}=1$, the model assumes that there are no outliers and that the measurements are Gaussian distributed; if $\zeta_{t}=0$, the outliers are considered and the correction step is not realized. In practice, the outliers might differently affect the elements in $y_{t}$, which motivates the use of the second solution.**Independent indicator VB Kalman filter (I-VKF)**: a complete vector $\zeta_{t}$ is considered, such that for each measurement an independent indicator is assigned [29,30]. Then, the model is
(35)$$p\left(y_{t}|x_{t},\zeta_{t}\right)=\prod_{i=1}^{n_{y}}N{\left(y_{t}^{\left(i\right)};h^{\left(i\right)}\left(x_{t}\right),{\left[R_{t}\right]}_{ii}\right)}^{\zeta_{t}^{\left(i\right)}}\;,$$
where $\zeta_{t}^{\left(i\right)}$ is an indicator for each code or carrier-phase observation. For the complete derivation and implementation details, refer to [29,30].

## 4. Validation and Experimentation

To evaluate the performance of the proposed robust RTK navigation algorithm, we considered first a synthetic scenario to statistically characterize the different methods and, then, we assessed their performance using real data from a measurement campaign.

### 4.1. Simulation Results

The duration of the synthetic scenario is 2000 s, with the observation step triggering at a 5 Hz rate (Δt=0.2 s). The rover moves following a constant velocity model and the uncertainty about the state transition is described by a white Gaussian random process $w_{t}∼N\left(0_{3},I_{3}\right)$, as described in Section 2.2. Indeed, the simulation setup resembles the example on target tracking in ([43] Ex. 16) for an LOS/NLOS environment, with the observations corresponding to the RTK model described in Section 2. Within the simulated trajectory, we defined two types of behaviours, a nominal one and a corrupted one (indicated as shaded gray areas in upcoming figures). In the corrupted grey zones and at every epoch, each satellite was randomly decided to be clean or an outlier, with a probability equal to 40% (α=0.40). If a satellite is resolved to be an outlier, the phase noise distribution does not change (i.e., it remains as in (37)) but it is affected by a cycle slip. Three different scenarios were considered based on the noise conditions of the observations. Recall that the *i*th satellite code and carrier phase pseudorange noises are $\varepsilon^{i}$ and $\epsilon^{i}$, respectively.

Case 0: Nominal. The noise distribution corresponds to the nominal case, for which
(36)$$\begin{array}{cc} \varepsilon^{i} & ∼N\left(0,{\left(\sigma_{\rho }^{i}\right)}^{2}\right),\;\sigma_{\rho }^{i}=0.5\;\cdot \;\left(1+1/\sin\left({\mathrm{elev}}^{i}\right)\right), \end{array}$$
(37)$$\begin{array}{cc} \epsilon^{i} & ∼N\left(0,{\left(\sigma_{\Phi }^{i}\right)}^{2}\right),\;\sigma_{\Phi }^{i}=\sigma_{\rho }^{i}/100. \end{array}$$
with ${\mathrm{elev}}^{i}$ as the ith satellite elevation.Case 1: Symmetric heavy-tailed. In this case, the noise distribution for the observations is described by
(38)$$\begin{array}{cc} \varepsilon^{i} & ∼\left(1-\alpha \right)N\left(0,{\left(\sigma_{\rho }^{i}\right)}^{2}\right)+\alpha N\left(0,{\left(100\;\cdot \;\sigma_{\rho }^{i}\right)}^{2}\right). \end{array}$$
which corresponds to a symmetric heavy-tailed noise scenario.Case 2: Skewed heavy-tailed. During the “corrupted” time periods, the noise distribution is as follows:
(39)$$\begin{array}{cc} \varepsilon^{i} & ∼\left(1-\alpha \right)N\left(0,{\left(\sigma_{\rho }^{i}\right)}^{2}\right)+\alpha N\left(10,{\left(10\;\cdot \;\sigma_{\rho }^{i}\right)}^{2}\right). \end{array}$$This corresponds to a skewed heavy-tailed noise scenario, where the second term has a positive mean and accounts for possible multipath conditions.

An example of the scalar noise distributions for the three scenarios is shown in Figure 3 (right plot). In addition, the left plot shows the considered skyplot, being the same for the three scenarios.

A total of seven filters were evaluated, which we can categorize as (a) conventional EKF, which include ideal EKFs (i.e., a filter with information on which observations are outliers and sets the best possible solution) and EKFs (i.e., a classical filter heavily affected by the presence of outliers); (b) robust statistics-based filters, which include RIFs based on Huber, Tukey, IGG, and 3σ score functions and GM-KF based on Huber score function; and (c) variational-based filters, which include scalar and independent indicator VFKs. The above-listed methods are described in Section 3, and the tuning parameters correspond to a=1.345 (Huber), c=4.685 (Tukey), and $k_{0}=1.5,k_{1}=2.5$ (IGG).

The evaluation metrics were (i) the root mean squared error (RMSE) for the float solution, i.e., how good the float estimation is; (ii) the mean ambiguity success rate (MASR), i.e., the percentage of Monte Carlo runs for which the vector of integer ambiguities are correctly estimated; and (iii) the cumulative distribution function (CDF) for the positioning errors during the float solution estimation. We opted for studying solely the precision for the float baseline positioning instead of the fixed solution, provided that the precision of fixed solution does not vary from method to method if the ambiguities are correctly estimated. Moreover, for case 0 (nominal case), the efficiency of the robust filters at the float estimation was addressed. The efficiency was defined as the relative performance of a particular estimator with respect to the optimal ML estimator (which corresponds to the EKF) under normal-distributed noise ([43] Chapter 1).

#### 4.1.1. Case 0: Nominal Gaussian Scenario

First, we considered the nominal Gaussian noise case. This is fundamental to assess the so-called efficiency or loss of efficiency in the different robust methods [51]. That is, when designing robust solutions, they are expected to perform close-to-optimal under nominal conditions, which is given by an efficiency close to 1. The results are shown in Figure 4. On the left, we show the relative efficiency obtained with the different methods, and on the right, we show the position RMSE for the float solution over time. The results on the percentage of successful ambiguity fixing is showcased in Table 2.

In these results, we can see the general good behavior of all the filters, with two interesting points: (i) the two variational-based robust KFs are the only ones that achieve an efficiency equal to 1 and (ii) the performances obtained with the RIF-IGG and GM-KF are slightly degraded w.r.t. the other filters. We also observe that all the filters manage to have the best possible success at ambiguity fixing, which in turn, grants a high precision for the fixed solution, regardless of their efficiency at the float estimation solution. However, posing good float estimates (i.e., having a high efficiency) remains a relevant factor to characterize the estimators, since it will condition the performance under more challenging scenarios, as we will observe next.

#### 4.1.2. Heavy-Tailed Noise Scenarios

Having assessed the performance of the different filters under nominal conditions, the next step is to analyze their behavior under the non-nominal heavy-tailed noise scenarios for cases 1 and 2. First, the float position RMSEs for both cases are shown in Figure 5. The EKF performance is significantly degraded when some satellites are affected by outliers (grey zones). The S-VKF presents the worst positioning performance among the estimators, the reason being that all observations are rejected during the gray areas, with the positioning solution being dictated by the prediction model. Thus, S-VKF might be interesting only whenever outliers are expected for very short time spans. On the contrary, the I-VKF allows for an almost ideal detection of the outliers, with a performance close to that of the Ideal EKF. This means that the burden of estimating the hyperparameters required for the variational inference does not penalize the overall performance. Indeed, the I-VKF showcases the overall best performance among the robust filters, both in terms of successful ambiguity fixing and RMSE for the float position estimates. For the skewed case 2, the performance obtained with the I-VKF is similar to the one provided by the RIF-Tukey and RIF-IGG. In both cases, the GM-KF and the RIF-Huber performances are slightly worse, and interestingly, the RIF with a simple 3σ rejection rule also performs quite well.

These performance results can also be seen in Figure 6, where we show the mean ambiguity success rate (MASR) over time. It is clear that, almost as soon as the NLOS/multipath periods start (shaded gray areas), the EKF becomes completely unable to successfully fix its ambiguities due to the influence of the outliers on its float estimates. The robust statistics-based filters present a higher resilience against such outliers, especially those with redescending functions (e.g., the RIF-Tukey, RIF-IGG, and RIF-3σ) since they can nullify the effect of the outliers. However, the detection capability offered by robust score functions is significantly worse than that of the variational-based filters. Naturally, the S-VKF is completely unable to fix any ambiguity during the contamination times, since all the observations are rejected and ambiguity cycle slips still occur during this time. On the other hand, I-VKF offers a performance almost identical to that of the ideal EKF for the case of symmetric heavy-tailed noises while being more sensitive to the skewed noise distributions.

To further complete these results, we show the empirical cumulative distribution function (CDF) in Figure 7 and the fix ratio (%) in Table 2. Again, we can see that the best solution both in terms of RMSE positioning performance and fix ratio is given by the I-VKF, which from this statistical analysis is the method of choice. However, this recommendation needs to be verified with real data in order to see the impact and mitigation capabilities in a real system. This analysis is provided in the sequel.

### 4.2. Real Data Experimentation

Performance characterization of the proposed robust RTK architecture is addressed for the navigation of a vessel navigation in an inland waterway channel. The measurement campaign was conducted in Koblenz (Germany) on 16 May 2017 (DOY 136, UTC 09:00–14:00), with the tracked vehicle being the MS Bingen, a multi-purpose research vessel of the German Waterway authorities. The scenario is shown in Figure 8, where the upper left plot depicts the number of satellites tracked as well as the position dilution of precision (PDOP). The trajectory followed by the vessel is illustrated in the bottom plot. Notice that the vessel performs several turns around the three consecutive bridges, which block the reception of satellite signals and induce multipath/NLOS conditions, leading to multiple cycle slips/losses in tracking.

The reference trajectory of the vessel was obtained based on optical technology, using a total station on land and an active reflector mounted under the GNSS antenna for automatic target tracking. This technology ensures a positioning accuracy of around one centimeter, for which the error pattern is independent from GNSS, and its availability is ensured even during maneuvers realized around bridges.

In this analysis, we considered the standard EKF compared solely to the RIF-Tukey, GM-KF, and I-VKF, which are the most promising robust techniques. The positioning error is shown in Figure 9 (left), and the corresponding CDF is shown in Figure 9 (right). First, notice that, in this scenario, the RIF-Tukey is not a good solution, with its performance being very close to the non-robust EKF. This is due to the low number of observations during bridge passing and the Huber function being monotone, meaning that the detected outliers are not fully eliminated from the estimation process. In contrast, both the I-VKF and GM-KF perform very well, with the I-VKF providing the lowest positioning error. This confirms the validity of the previous synthetic data results, where the I-VKF performance is always superior. However, it is also worth pointing out that the GM-KF is not a solution to be disregarded in practice in contrast with the previous results in Section 4.1.2. Indeed, the GM-KF is the only method that accounts for possible errors in the KF prediction stage, and therefore, using real dynamics that may not exactly follow the assumed dynamic model, this may be a critical point. For completeness, the fix ratio is given in Table 3, where we can see that the ratios are low compared to the synthetic data scenario. This is due to two main reasons: (i) even if the float solution is good using the robust methods, there are not enough observations to ensure a high fix ratio, and (ii) the bridges induce multiple cycle slips, which degrade the system observability.

## 5. Conclusions

It is well known that GNSSs play a key role in a plethora of intelligent transportation system applications, where reliable precise positioning solutions are fundamental. A great challenge for precise carrier phase-based positioning techniques, such as PPP and RTK, relates to their applicability in harsh propagation conditions. Indeed, a key step of these techniques is the phase ambiguity resolution, which is particularly affected by such harsh conditions. Thus, new robust filtering solutions must be developed to enable the use of RTK in scenarios involving non-nominal measurement errors, such as multipath in urban canyons. In this contribution, we explored the use of robust filtering techniques to mitigate the impact of outliers. The driving idea is to replace the first stage of a standard EKF of the RTK pipeline with a robust filter to obtain a robustified float solution. Kalman filtering solutions based on the frameworks of robust statistics and variational inference are presented, and some basic theoretical concepts from both schemes are introduced. Different robust filters to replace the original RTK float estimation were proposed, namely the RIF based on distinct score functions, the KM-KF, or the Independent indicator VKF. It was shown, with both synthetic and real data, that using such a robust RTK architecture significantly improves the overall system performance, being a promising solution for precise positioning in harsh environments. Among the different robust filters available in the literature, the I-VKF was found to provide the overall best performance. The GM-KF is also worthy of an honorable mention, allowing coping with inaccurate dynamical models, which is an interesting feature in scenarios with changing dynamics.

## Figures and Tables

**Figure 1 sensors-21-01250-f001:**
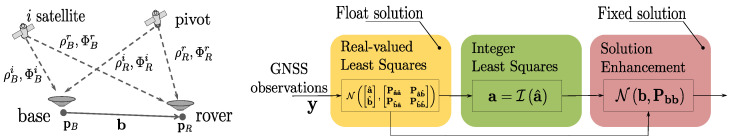
On the (**left**), a scheme for the real-time kinematic (RTK) procedure. On the (**right**), the workflow for the ambiguity resolution process.

**Figure 2 sensors-21-01250-f002:**
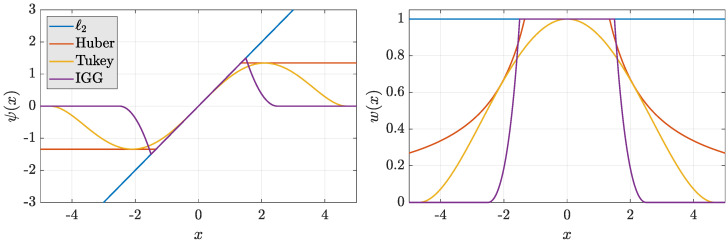
Illustration of the score (**left**) and weighting (**right**) functions for conventional LS ($\ell_{2}$) and robust estimators: here, the families of Huber, Tukey, and IGG functions are depicted with parameters a=1.345 and c=4.685, $k_{0}=1.5$, and $k_{1}=2.5$ respectively.

**Figure 3 sensors-21-01250-f003:**
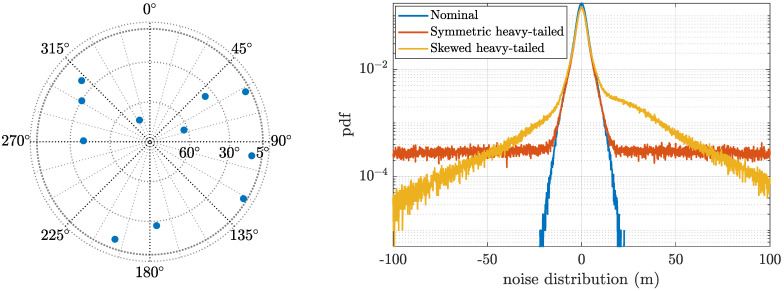
Skyplot and noise distributions for the different scenarios.

**Figure 4 sensors-21-01250-f004:**
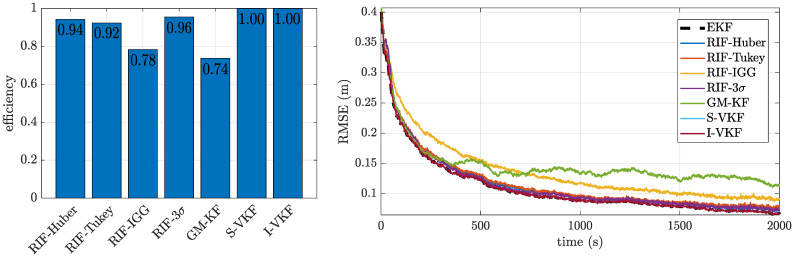
Efficiency and errors over time for the nominal Gaussian-distributed noise of case 0.

**Figure 5 sensors-21-01250-f005:**
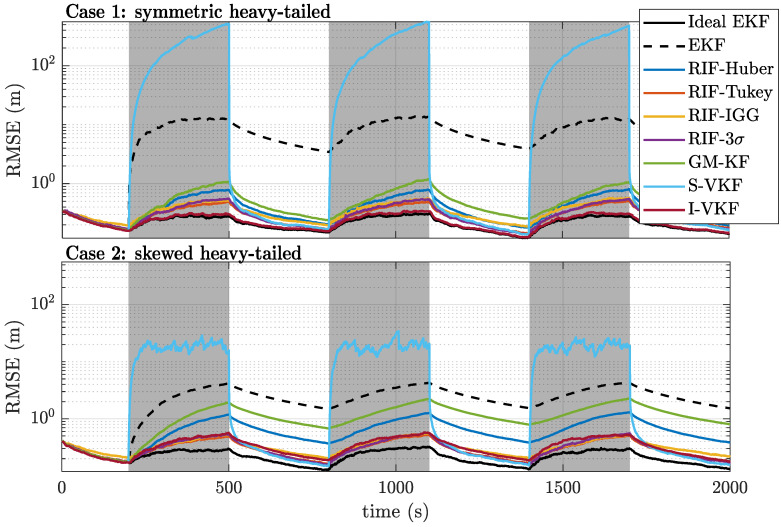
Root mean squared error (RMSE) for the float positioning results over time for cases 1 (**top**) and 2 (**bottom**).

**Figure 6 sensors-21-01250-f006:**
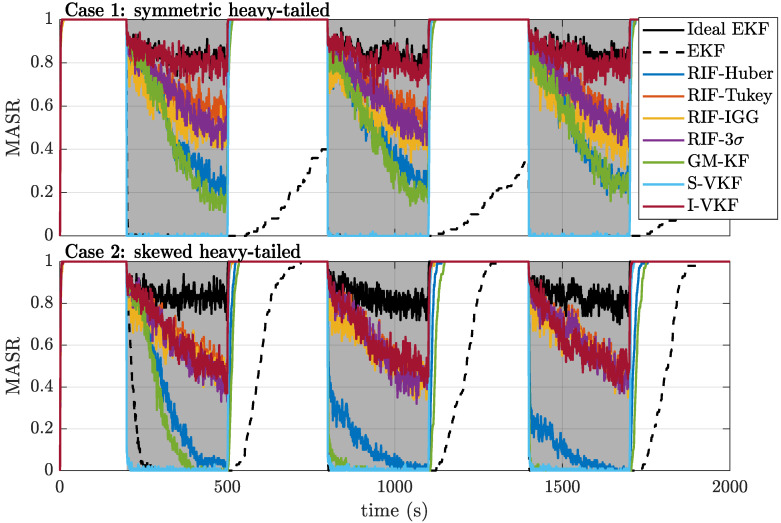
Mean ambiguity success rate (MASR) over time for cases 1 (**top**) and 2 (**bottom**).

**Figure 7 sensors-21-01250-f007:**
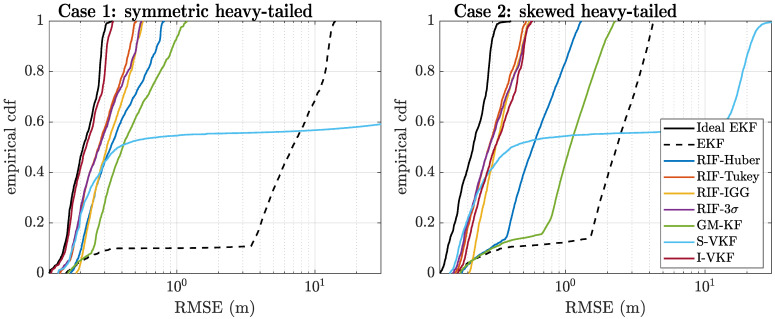
Empirical cumulative distribution function (CDF) of the float positioning errors for cases 1 (**left**) and 2 (**right**).

**Figure 8 sensors-21-01250-f008:**
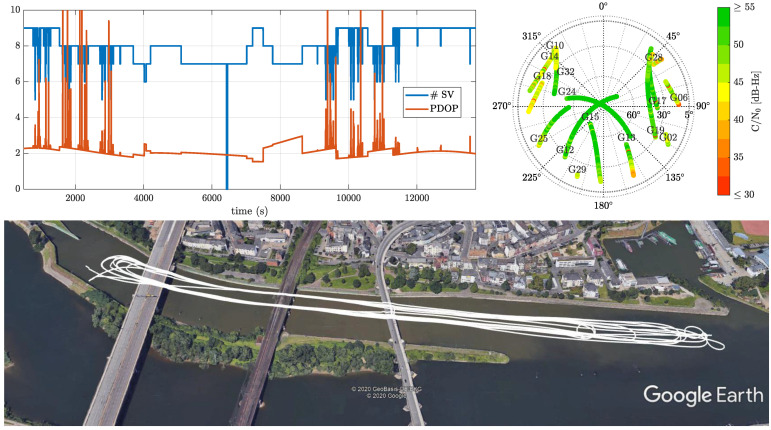
On the upper left, the number of tracked satellites and associated position dilution of precision (PDOP). On the upper right, a skyplot of the scenario, with the colorbar indicating the C/N${}_{0}$. At the bottom, the trajectory followed by the tracked vessel during bridge passing, estimated using laser technology.

**Figure 9 sensors-21-01250-f009:**
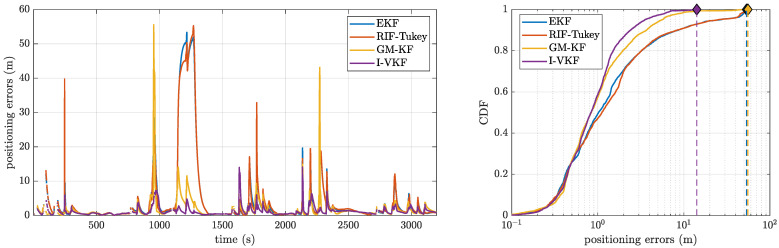
(**Left**) Positioning errors; (**right**) cumulative distribution function (CDF) for the different methods.

**Table 1 sensors-21-01250-t001:** Overview on robust estimation for global navigation satellite system (GNSS) applications.

Framework	Processing stage	Application	References
**Robust Statistics**	Baseband processing	Interference mitigation	[11,12,13,14]
	PVT processing	Snapshot code-based positioning	[15,16,17,18,19]
		Recursive code-based positioning	[20,21,22]
		Recursive RTK/PPP	[23,24]
**Variational Inference**	Baseband processing	Adaptive phase tracking	[25,26]
	PVT processing	Recursive code-based positioning	[27,28]
		Recursive RTK/PPP	[29,30]

**Table 2 sensors-21-01250-t002:** Percentage of successful fixed solutions (%).

	Conventional EKF		Robust Statistics-Based					Variational-Based	
	Ideal EKF	EKF	RIF-Huber	RIF-Tukey	RIF-IGG	RIF-3σ	GM-KF	S-VKF	I-VKF
**Case 0**	99.96	99.96	99.96	99.96	99.96	99.96	99.96	99.96	99.96
**Case 1**	92.74	16.20	76.63	85.42	80.41	84.19	75.38	54.46	91.82
**Case 2**	92.47	39.89	61.42	83.62	81.47	82.45	56.31	54.60	83.22

**Table 3 sensors-21-01250-t003:** Percentage of fixed solutions (%).

	Estimator			
	EKF	RIF-Tukey	GM-KF	I-VKF
Fix ratio (%)	53.22	46.74	58.27	53.46

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
