# Peer review of "Robust Filtering Techniques for RTK Positioning in Harsh Propagation Environments"

_sensors, 2021, doi:10.3390/s21041250_

Round 1

Reviewer 1 Report

This manuscript proposes robust filtering techniques to mitigate the impact of outliers for RTK. More Specially, the standard EKF is replaced by a robust filter. The effectiveness of the proposed algorithm is verified by the synthetic and real data. This manuscript is well written and organized in its current form. The experimental results are sufficient. However, there are still typeset errors:
(1) In Figure 2, illustration of score is missed which is stated at the title of Figure 2 as "score (middle)".
(2) In line 223, ...already shown in section ??...
(3) In line 171, ...further extended and analyzed in [27?]...

Author Response

We sincerely appreciate the comments received by the reviewer. Also, we apologize for the typos present in the first version of the manuscript. We have extensively inspect the paper for its reviewed version.

Reviewer 2 Report

Robust filters were used in the RTK resolution and verified. I would like to recommend the manuscript be accepted for publication after revisions. Specific comments are listed as follows:

  1. Figure 1: The caption should be "On the left,...".
  2. L120: The authors should add some references for the application of KF, EKF, and SPGF to RTK resolution or typical RTK programs.
  3. L132 and 223: Some descriptions are not completed, e.g., "Sect. xx-" and "Section ??". The authors should revise these two errors and check other similar errors.
  4. Fig. 4, Fig. 6, L236, and Table 1: "Efficiency", "MASR", "fix ratio", and "percentage of fixed solutions" are used. If they represent the same conception, they should be unified. If not, the authors should tell the differences.
  5. Fig. 4 left and Table 1: The data of efficiency in left Fig. 4 are not consistent with the data of percentage of fixed solutions in Table 1.
  6. Figure 4: It's not easy to identify some lines.
  7. Figs. 4-6: The horizontal axis title could not represent the elapsed time. Is it "time(s)"?
  8. Table 1: What's the functionality of the horizontal dashed lines?
  9. Case numbers should be unified; some use the Arabic numerals but some use Roman numerals.
  10. Figure 7 should be placed in Section 4.1.
  11. L237-238: According to Figs. 5-7 and Table 1, the ideal EKF seems to be better than the I-VKF. Thus, the I-VKF is fine but it isn't the best.
  12. Fig. 9 and Table 2: Why the other four filters, including RIF-Huber, RIF-IGG, RIF-3 sigma, and S-VKF, are not validated?

Author Response

We thank the reviewer for his/her insightful comments. Please, see the attached PDF for the detailed response. 

Reviewer 3 Report

Dear Editor,
my comments for authors:
1. The obtained results and findings from research must be included in Abstract. Currently, I don't see it.

2. all acronyms must be explained in the text, please check it.

3. In introduction you write about ambiquity solution. But I don't see any information new solution of this parameter, e.g. PREFMAR method. In my opinion the Introduction must be revised, please add few papers about PREFMAR method for ambiguity estimation.

4. algorithm (1-40), I don't see the reference to equations? It means that all equations are your property? please explain it.

5. chapter simulation results, please clearly write the value of a priori of code and phase observation errors.

6. all symbols in equations (1-40) must be explained in the text.

7. conclusions, please underline what was the novelty of paper and please write your findings from research test.

8. Please modify the References list

Author Response

We appreciate the reviewer for his/her comments. We believe that your observations have sincerely contributed to improve our manuscript. Please see the attached PDF for detailed replies.

Round 2

Reviewer 3 Report

I accept the paper.